# Analysis of the Flow in a Typified USBR II Stilling Basin through a Numerical and Physical Modeling Approach

**Juan Francisco Macián-Pérez** [1,*] **, Rafael García-Bartual** [1] **, Boris Huber** [2] **, Arnau Bayon** [1] **and Francisco José Vallés-Morán** [1]

1 Research Institute of Water and Environmental Engineering, Universitat Politècnica de València, Camí de Vera, s/n, 46022 València, Spain; rgarciab@hma.upv.es (R.G.-B.); arbabar@iiama.upv.es (A.B.); fvalmo@hma.upv.es (F.J.V.-M.)
2 Institute of Hydraulic Engineering and Water Resources Management, Technische Universität Wien, Karlsplatz 13/222, A-1040 Vienna, Austria; boris.huber@tuwien.ac.at
* Correspondence: juamapre@cam.upv.es; Tel.: +34-963-877-613

**Abstract:** Adaptation of stilling basins to higher discharges than those considered for their design implies deep knowledge of the flow developed in these structures. To this end, the hydraulic jump occurring in a typified United States Bureau of Reclamation Type II (USBR II) stilling basin was analyzed using a numerical and experimental modeling approach. A reduced-scale physical model to conduct an experimental campaign was built and a numerical computational fluid dynamics (CFD) model was prepared to carry out the corresponding simulations. Both models were able to successfully reproduce the case study in terms of hydraulic jump shape, velocity profiles, and pressure distributions. The analysis revealed not only similarities to the flow in classical hydraulic jumps but also the influence of the energy dissipation devices existing in the stilling basin, all in good agreement with bibliographical information, despite some slight differences. Furthermore, the void fraction distribution was analyzed, showing satisfactory performance of the physical model, although the numerical approach presented some limitations to adequately represent the flow aeration mechanisms, which are discussed herein. Overall, the presented modeling approach can be considered as a useful tool to address the analysis of free surface flows occurring in stilling basins.

**Keywords:** USBR II stilling basin; hydraulic jump; physical model; numerical model; void fraction; CFD

## 1. Introduction

The crucial role played by dams in civil engineering can only be understood due to its significant economic and social importance, which leads to high safety requirements, as a result of the critical consequences derived from a possible failure. This singularity, bound to the important growth in the number of dams built all around the world during the last decades, brings the importance of dam engineering into the spotlight [1].

Dams need to deal with the evacuation of excess water coming from floods. The parts of the dam intended for this purpose are the spillways, which are usually complemented by stilling basins, to ensure the restitution of water to the river with the appropriate energy conditions. The flow taking place in these structures is particularly complex, as a result of its multiphase nature and the large turbulence developed [2,3], which needs to be considered during the design process. This design has traditionally been approached from the perspective of experimental campaigns, using reduced

physical models which, despite their utility, introduce some unavoidable scale effects that may affect some aspects of the flow, such as the aeration [4,5]. Alternatively, computational fluid dynamics (CFD) techniques arise as advanced tools providing a detailed analysis of the flow at the prototype scale. This approach is gaining more and more importance in the modeling process of hydraulic structures, especially those where aeration needs to be considered [4,6]. In spite of the multiple advantages of CFD techniques, they require a number of hypotheses and theoretical simplifications, and thus, calibration and validation using physical models remain indispensable [7,8]. It is this complementary nature of numerical and experimental techniques that motivates the double modeling approach carried out in the present research.

On this basis, both, a numerical and a physical model of a spillway and a United States Bureau of Reclamation Type II (USBR II) stilling basin were developed. The objective of the research was to study the flow taking place in stilling basins designed for energy dissipation purposes. A better understanding of this flow is crucial, since dam adaptation to new standards derived from society demands regarding flood protection and climate change effects remains a challenge, especially for the energy dissipation structures [9]. Hence, the results of this research were intended to enhance the knowledge of the flow taking place in stilling basins, in order to improve their performance and to contribute to the adaptation of existing dams to higher discharges than those originally considered in their design.

The flow in energy dissipation structures and, in particular, the hydraulic jump, which is the hydraulic phenomenon generally forced in stilling basins to dissipate energy, has been widely studied [3,10–13]. The hydraulic jump constitutes an extremely complex phenomenon, due to its characteristics, involving turbulence, air entrainment, and pressure and velocity fluctuations. In this respect, the free-surface profile of the hydraulic jump was investigated by Chachereau and Chanson and Zhang et al. [14,15], who studied how pressure and velocity fluctuations affect the hydraulic jump shape, whereas Wang and Chanson [8] focused on the hydraulic jump toe position. Furthermore, Mossa [16] investigated the oscillating characteristics of hydraulic jumps and their relation with pressure and velocity fluctuations. Regarding air entrainment, Chanson and Brattberg and Murzyn et al. [17,18] studied the void fraction distribution in the hydraulic jump. Moreover, Gualtieri and Chanson [19] addressed the effect of the inflow Froude number on air entrainment in the hydraulic jump, complemented by a study on similitude and scale effects of this process [20]. However, despite the intense research effort devoted to improving knowledge and modeling of the hydraulic jump, its complexity has prevented a full understanding of the phenomenon [3,8].

Taking into account the above-mentioned inherent complexity, a double modeling approach was adopted in the present research. On the one hand, the CFD code FLOW-3D® was used to model the spillway and stilling basin case study. This commercial software, widely used for hydraulic engineering, has proved to successfully reproduce hydraulic jumps as well as the flow taking place in spillways and energy dissipation structures [3,21–23]. On the other hand, a reduced-scale physical model with Froude similarity of the case study was built in the Hydraulics Laboratory of the Institute of Hydraulic Engineering and Water Resources Management (Technische Universität Wien, Wien, Austria), following the limiting criteria to avoid significant scale effects proposed by Heller [5].

The case study selected to undertake this double modeling approach was based on the analysis of a series of existing dams in the Júcar River Basin (Spain), prioritizing the design of a general and representative case. From this analysis, the design of the structure was established and the dimensions and discharge of the spillway and stilling basin were determined (Figure 1). Regarding the spillway, a Creager profile was designed [24], altogether with the calculations devoted to obtaining the reservoir water level that leads to the established discharge [1,24–26]. For the energy dissipation structure, a typified USBR II stilling basin was designed, following the recommendations and patterns of the United States Bureau of Reclamation [27], which allow obtaining not only the dimensions of the basin but also the size and distribution of the energy dissipation devices. The reason to choose a Creager

profile spillway and a typified USBR II stilling basin is that both have been widely used and studied all around the world [28–30] and thus, are considered to constitute a representative case study.

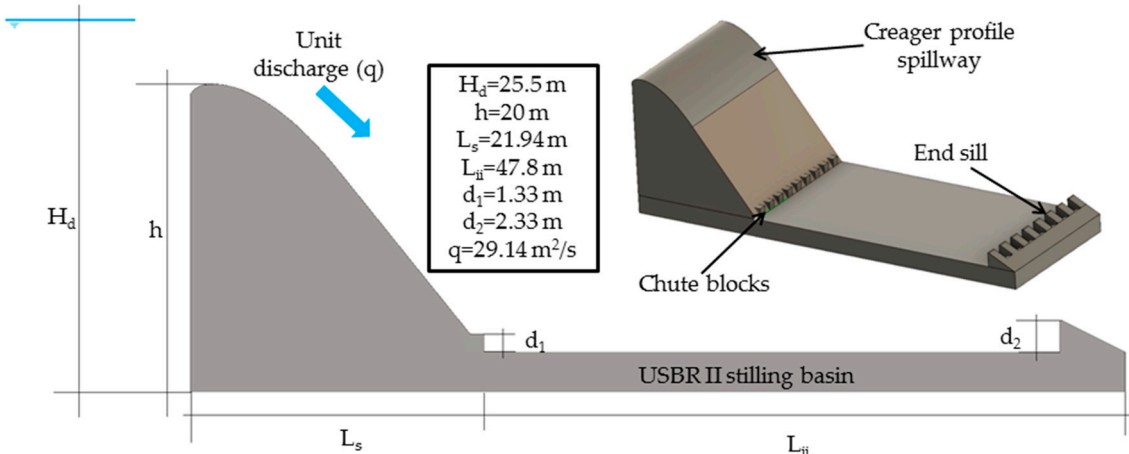

**Figure 1.** Case study: Creager profile spillway and United States Bureau of Reclamation Type II (USBR) type II stilling basin with the basic information regarding flow conditions and dimensions.

Apart from the dimensions shown in Figure 1, it is important to remark that the width and spacing of the chute blocks equal its height ($d_1$ = 1.33 m) whereas, for the end sill, the width and spacing of the blocks is 1.67 m.

Once the case study was defined and modeled with the previously referred double approach (numerical and physical), the research herein presented seeks to improve the performance of the energy dissipation structure through a better understanding of the flow occurring within the stilling basin. To do so, a series of features and characteristics of the hydraulic jump developed in the basin were studied and compared, not only between the numerical and physical models but also using results presented by other authors, coming from an extensive literature review. On this basis, the goal of the research is to present combined numerical and physical modeling as a tool to reach a better understanding of the flow taking place in hydraulic engineering structures, which, in turn, can be useful to improve their design in order to tackle adaptation challenges derived from new security standards and climate change effects.

Hence, a series of structural properties regarding the hydraulic jump taking place in the USBR II stilling basin were analyzed in this paper. Firstly, in terms of the hydraulic jump shape, the free surface profile and the sequent depth ratio were studied, altogether with the hydraulic jump efficiency. Furthermore, values regarding velocity profiles in different positions and pressure in the energy dissipation devices existing in the basin were analyzed. Finally, the void fraction distribution in a series of vertical profiles along the stilling basin was obtained and discussed.

## 2. Materials and Methods

### 2.1. Numerical Model

The numerical model developed to analyze the presented case study uses the commercial CFD code FLOW-3D® [31]. In particular, the prototype scale dimensions were considered for the numerical model, whose characteristics are presented in the forthcoming subsections.

#### 2.1.1. Flow Equations and General Settings

FLOW-3D bases its results on the flow governing equations, namely the Navier–Stokes equations (Equations (1) and (2)). Due to the characteristics of the analyzed flow, the Navier–Stokes equations for incompressible fluids were employed. It is important to highlight that for the numerical resolution of

the equations, FLOW-3D uses the finite volume method [32] in order to discretize the conservation laws in the case study spatial domain.

$$\nabla = 0 \tag{1}$$

$$\frac{\partial \overline{u}}{\partial t} + \overline{u} \times \nabla \overline{u} = -\frac{1}{\rho} \nabla p + \nu \nabla^2 \overline{u} + \overline{f_b} \tag{2}$$

where $u$ is the velocity, $t$ is the time, $\rho$ is the fluid density, $p$ the pressure, $\nu$ the fluid kinematic viscosity and $f_b$ accounts for the body forces (i.e., gravity and surface tension).

Regarding time discretization, the time-step size was automatically adjusted by the code, using a Courant-type stability criterion to improve model efficiency with a reduction of computational times and to minimize numerical divergence risk [31].

### 2.1.2. Free Surface Modeling

FLOW-3D bases its strategy to model and track the free surface on the volume of fluid (VOF) method [33]. Hence, a variable named fraction of fluid ($F$) was used to determine the fractional volume of the main fluid (i.e., water in the presented case), so that it reaches a value of 1 when the cell is completely filled with water and a value of 0 when it is empty. In order to track the evolution of the fraction of fluid throughout the domain, the following expression was used:

$$\frac{\partial F}{\partial t} + \nabla \times (\overline{u}F) = 0 \tag{3}$$

Furthermore, in the treatment of hydraulic problems involving a free surface between air and water, FLOW-3D allows one fluid approach for the resolution of the flow governing equations. With this approach, the boundary conditions are applied to the free surface in order to solve the equations only for the water phase, whereas the gas is assumed to have negligible inertia and only applies normal pressure on the free surface [34]. Therefore, there is a significant reduction in computing times.

### 2.1.3. Turbulence Modeling

The CFD model solves the flow governing equations using the Reynolds averaging of the Navier–Stokes equations (RANS), which is the most extended approach in engineering applications, due to computing time limitations [3]. Averaging the Navier–Stokes equations leads to the appearance of the Reynolds stresses in the analysis, and the addition of new variables related to the turbulent viscosity. This approach leads to the known closure problem in the flow governing equations, which can be tackled through a convenient turbulence model. There are different types of turbulence models according to the number of equations employed to solve the closure problem. Among these models, two-equation turbulence models, which use two transport equations for variables related with the turbulent viscosity, are the most frequent option since they are able to provide a reliable description of turbulence in terms of time and space scales [35].

Among the two-equation models, three of the most extended options were tested for the present study. Firstly, the $k$-$\varepsilon$ model [36,37] was tested. This model involves two transport equations, one for the turbulent kinetic energy ($k$) and another one for its dissipation rate ($\varepsilon$). The $k$-$\varepsilon$ model has shown good performance for a wide range of flows [38]. The second model tested was the RNG (renormalization-group) $k$-$\varepsilon$ [39]. It applies statistical methods to derivate the averaged equations for the turbulence quantities employed by the $k$-$\varepsilon$ model, showing a better ability to represent flows with strong shear effects [23]. Finally, the $k$-$\omega$ turbulence model was employed [40]. Under certain conditions, this model usually provides reliable approximations for specific flow conditions, such as flow near wall boundaries or with streamwise pressure gradients [31].

Hence, the case study was simulated using the above mentioned two-equation turbulence models. The results of the numerical simulations allowed quantifying and describing in detail certain relevant hydraulic jump characteristics, such as the free surface profile, sequent depth ratio, and hydraulic jump

efficiency. All of them were analyzed and compared with previous results coming from a bibliographical review [41]. This comparison showed that all of the models were able to successfully reproduce the case study in terms of the chosen variables. However, and despite the similar overall performance of the three turbulence models, the RNG *k-ε* led to results closer to those previously documented for a typified USBR II stilling basin. Furthermore, this turbulence model has been successfully used in hydraulic structures numerical modeling in recent years [42,43].

Accordingly, this was the turbulence model selected for the present work. In the RNG *k-ε* model, the transport of the turbulent kinetic energy and its dissipation rate are modeled by the equations:

$$\frac{\partial}{\partial t}(\rho k) + \frac{\partial}{\partial x_i}(\rho k u_i) = \frac{\partial}{\partial x_j}\left[\left(\mu + \frac{\mu_t}{\sigma_k}\right)\frac{\partial k}{\partial x_j}\right] + P_k - \rho\varepsilon \tag{4}$$

$$\frac{\partial}{\partial t}(\rho\varepsilon) + \frac{\partial}{\partial x_i}(\rho\varepsilon u_i) = \frac{\partial}{\partial x_j}\left[\left(\mu + \frac{\mu_t}{\sigma_\varepsilon}\right)\frac{\partial\varepsilon}{\partial x_j}\right] + C_{1\varepsilon}\frac{\varepsilon}{k}P_k - C_{2\varepsilon}\rho\frac{\varepsilon^2}{k} \tag{5}$$

where $x_i$ is the coordinate in the *i* axis, $\mu$ is the dynamic viscosity, $\mu_t$ is the turbulent dynamic viscosity and $P_k$ is the production of turbulent kinetic energy. Finally, the value for the parameters $\sigma_k$, $\sigma_\varepsilon$, $C_{1\varepsilon}$ and $C_{2\varepsilon}$ is given by Yakhot et al. [39].

### 2.1.4. Air Entrainment

Flow aeration constitutes a crucial feature in hydraulic jumps, due to the occurrence of shear layers, eddies, and free surface fluctuations causing air entrapment, which, in turn, influences the flow characteristics [6,44,45]. Hence, accurately modeling air entrainment and its affection to flow constitutes a key issue in the simulation of hydraulic jumps. However, the presence of air bubbles and droplets, which may present a characteristic length below the mesh size, adds considerable complexity to the simulation of the phenomenon [3,46].

FLOW-3D models air entrainment through a balance between stabilizing forces (gravity and surface tension) and destabilizing forces (turbulent kinetic energy), so that an estimation of the air entrainment rate to the flow is carried out. In these terms, the volume of entrained air rate ($\delta V$) is obtained as [23]:

$$\delta V = k_{air}A_S\left[\frac{2(P_t - P_d)}{\rho}\right]^{\frac{1}{2}} \; if \; P_t > P_d; \; \delta V = 0 \; if \; P_t < P_d \tag{6}$$

$$P_t = \rho\rho k; \; P_d = \rho g L + \frac{\sigma}{L_T} \tag{7}$$

$$L = \frac{C_\mu^{\frac{3}{4}} k^{\frac{3}{2}}}{\varepsilon} \tag{8}$$

where $P_t$ and $P_d$ are the destabilizing and stabilizing forces, respectively, $L$ is the turbulence length scale and $C_\mu$ has a constant value of 0.085 when using the RNG *k-ε* turbulence model. Furthermore, $\rho$ is the water density, $g$ the gravity component perpendicular to the water surface, $\sigma$ the surface tension coefficient, $k_{air}$ is a coefficient of proportionality that must be specifically calibrated for each case and $A_S$ is the free surface area.

For cases in which aeration constitutes an important feature, affecting the behavior of the flow, such as the one presented here, it is necessary to consider additional physical processes of air transport in water [23]. To do so, FLOW-3D takes into account bulking and buoyancy effects by using the models presented hereafter [31].

On the one hand, the density evaluation model considers the varying fluid density resulting from air entrainment, computing the fluid mixture density ($\rho_m$) in each cell as a linear relationship of the two fluid densities, i.e., the water density ($\rho$) and the air density ($\rho_a$):

$$\rho_m = F\rho + (1-F)\rho_a \tag{9}$$

On the other hand, the drift-flux model accounts for the interaction between the two phases, with the air bubbles moving within the fluid as a consequence of the difference in densities and thus affecting the fluid motion. Hence, in the calculation of the drag between phases, the drift-flux model obtains the drag per unit volume ($K_P$) as:

$$K_P = \frac{1}{2} A_P \rho \left( C_D U + 12 \frac{\mu}{\rho R_P} \right) \tag{10}$$

where $A_P$ is the cross-sectional area per unit volume of the dispersed phase (i.e., air), $C_D$ is a drag coefficient defined by the user, being 0.5 the general default value for spheres, $U$ is the magnitude of the relative/slip velocity, $\mu$ the water dynamic viscosity and $R_P$ the average particle radius. Furthermore, the minimum and maximum volume fraction values for water were established as 0.1 and 1, respectively, allowing gas to escape at the free surface. This implies that the free surface is identified with an air concentration of 90%, as commonly indicated in experimental research [23]. Regarding the volume fraction threshold that controls when the dispersed phase turns into a continuous fluid, it was set so that water always remains as the continuous phase, as recommended by FLOW-3D [31]. In addition to this, the potential affection of high air fractions to the relative velocity between phases ($u_r$) is considered adopting the Richardson–Zaki approach [47], which introduces an adjusted relative velocity ($u\prime_r$):

$$u\prime_r = u_r \times \max(0.5; F)^{R_m R_z} \tag{11}$$

where $R_m$ is the Richardson–Zaki coefficient multiplier and $R_z$ the Richardson–Zaki coefficient, determined from the bubble Reynolds number.

### 2.1.5. Meshing and Boundary Conditions

The spatial domain subject of the present work was meshed using a structured rectangular hexahedral mesh with two different mesh blocks. Hence, a containing mesh block was created for the entire spatial domain, and then, a nested mesh block was built, with refined cells for the area of interest, where the hydraulic jump takes place. Furthermore, areas where flow is not expected were cropped in the meshing process to increase the efficiency of the simulation without affecting the results (Figure 2).

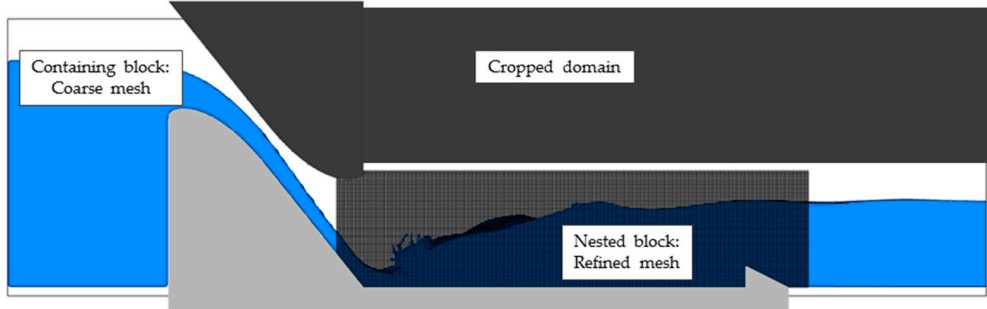

**Figure 2.** Meshed domain and detail of the containing and the nested mesh blocks.

For the determination of the cell size, a mesh convergence analysis was carried out in order to achieve the independence of the model results from the imposed cell size. The analysis was developed following the Richardson extrapolation method [48]. To do so, four different meshes were tested, choosing ten basic variables (i.e., velocities and pressures) per mesh as indicators. Each mesh had different cell sizes, ranging from 0.8 to 0.135 m, which accomplished the minimum refinement ratio of 1.3 recommended by Celik et al. [48]. Once the mesh convergence analysis was performed, the mesh consisting of a containing block with a cell size of 0.36 m and a nested block of 0.18 m was chosen. The resulting model apparent order was 2.78, slightly above the model formal order. The grid convergence index (GCI) was around 6%, which can be considered as an acceptable value for cases

involving complex flows such as the one here studied [3]. Tables 1 and 2 show a summary of the results of the mesh convergence analysis conducted.

**Table 1.** Characteristics of the meshes tested in the convergence analysis.

| Mesh | Nested Block Cell Size | Containing Block Cell Size |
|:---:|:---:|:---:|
| 1 | 0.400 m | 0.800 m |
| 2 | 0.250 m | 0.500 m |
| 3 | 0.180 m | 0.360 m |
| 4 | 0.135 m | 0.270 m |

**Table 2.** Results of the mesh convergence analysis.

| Mesh Combination | Model Apparent Order (p) | Grid Convergence Index (GCI) |
|:---:|:---:|:---:|
| 1-2-3 | 2.78 | 6.0% |
| 2-3-4 | 5.13 | 6.4% |
| 1-3-4 | 133.23 | 3.6% |
| 1-2-4 | 2.51 | 13.6% |

Regarding the boundary conditions, at the upstream boundary, a volume flow rate with the corresponding fluid elevation was set, according to the designed case conditions. At the downstream boundary, fluid leaves the domain with an imposed flow depth so that the hydraulic jump occurs in the correct location, as established in the selected case study. Reaching such conditions under the CFD modeling approach is not a straightforward process. A sequence of successive runs was needed, each of them reproducing a non-stationary flow. Finally, a simulation that satisfactorily represents a stable hydraulic jump in the stilling basin was obtained, with the pre-selected discharge and an adequate mesh resolution, as explained above. This final simulation achieves a steady state for the flow, presenting a variation of the fluid fraction in the domain under 2%. Once such a condition was reached, additional 10 s of simulation were used to collect and average the variables, which are analyzed in the forthcoming sections.

For the top boundary, atmospheric pressure was set, whereas for the bottom, a wall non-slip condition was established with a high Reynolds number wall function imposed on the solid contours.

*2.2. Physical Model*

The reduced scale physical model for the case study was built in the Hydraulics Laboratory of the Institute of Hydraulic Engineering and Water Resources Management, at the TUWien. Its design was made considering the scale effects limiting criteria stated by Heller [5], altogether with the available resources at the Hydraulics Laboratory. Consequently, a scale factor of 1:25 was adopted, with regards to the dimensions displayed for the prototype case study shown in Figure 1. In terms of the model construction, a rectangular section open flow channel was used to locate the spillway and the stilling basin. This channel is equipped with a downstream gate, that can be operated to achieve the hydraulic jump desired position and a glass wall in the area of interest (Figure 3).

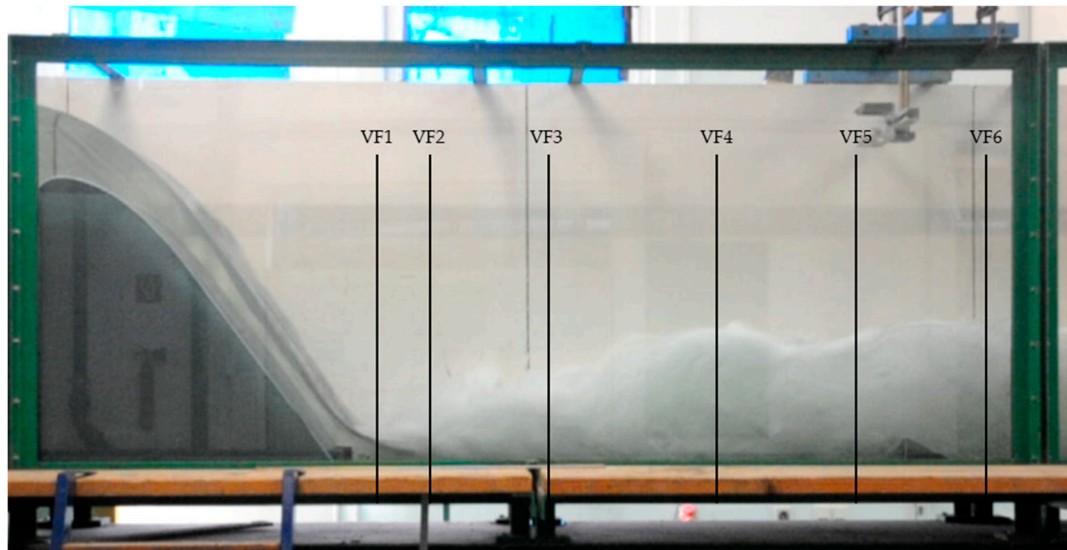

**Figure 3.** Physical model of the case study at the TUWien Hydraulics Laboratory with the profiles used to measure the void fraction in the experimental campaign (VF1–VF6).

The experimental campaign carried out characterized the hydraulic jump through a series of features, measured with the corresponding equipment. Hence, flow depths were measured to obtain the free surface profile, the sequent depth ratio, and the hydraulic jump efficiency. Moreover, velocities in a series of vertical profiles and pressures in the end sill of the stilling basin were measured. Finally, the void fraction distribution was obtained in several vertical profiles along the hydraulic jump longitudinal axis (Figure 3). It is important to highlight that the flow under study is extremely complex. In this regard, it should be emphasized that obtaining reliable measures of certain variables remains a challenging goal, given the available measuring devices and experimental limitations. Consequently, there is an unavoidable degree of uncertainty associated with the parameters studied in the experimental campaign. However, the design of the campaign was made trying to reduce as much as possible this uncertainty, choosing adequate measuring times and locations. Furthermore, a preliminary analysis of the data was made in order to discard anomalous values, as stated in forthcoming sections.

### 2.2.1. Digital Image Processing

The free surface longitudinal profile of the hydraulic jump taking place in the physical model was obtained using Digital Image Processing (DIP) techniques. To do so, a video of the model from its side was recorded, during 30 s and with an acquisition frequency of 25 Hz (fps). An edge detection method based on a light intensity threshold was applied to identify air–water interfaces [3]. To enhance the reliability of this method, point measures with limnimeters were taken for calibration purposes. Furthermore, perspective effect correction and filtering algorithms to remove the bias caused by droplets, reflections, and others were applied.

### 2.2.2. Turbine Velocity Meter

Streamwise velocity profiles were obtained in different positions within the physical model using a turbine velocity meter by Schiltknecht (MiniWater20®). The working principle of this device is based on a turbine whose rotation frequency depends on the water velocity. Then, this rotation frequency is converted into an analog output that gives a measure of the flow velocity. The velocity profiles obtained in the physical model covered the following positions: the flow in the spillway, right before the entrance to the stilling basin, the flow in the stilling basin, close to the end sill, and the flow downstream of the end sill. Measures with the turbine velocity meter were taken during 180 s for each

of the points forming the profiles, with an acquisition frequency of 2 Hz. The relatively low acquisition frequency of the device prevented taking velocity measures in the hydraulic jump roller, where velocity fluctuations are more intense.

### 2.2.3. Pressure Transmitters

The pressure in two different positions of the end sill was measured with pressure transmitters, which turn the pressure in their sensors into an electric signal with their piezoresistive transducer and microprocessor with a converter. The pressure measures were taken during 60 s with an acquisition frequency of 50 and 200 Hz, showing that the first of the frequencies was sufficient to ensure stable results.

### 2.2.4. Optical Fiber Probe

The void fraction distribution within the hydraulic jump was obtained by measuring with a dual-tip optical fiber probe at six different profiles along the hydraulic jump longitudinal axis. Such profiles are shown in Figure 3. For each one of them, 9 to 12 points were measured. The collection data time for each of the points was 200 s, due to the characteristics and expected velocities in the model [18].

The optical fiber probe employed in the experimental campaign was an RBI Instrumentation© dual-tip optical phase detection device, which works on the basis presented by Cartellier and Archard [49], Cartellier and Barrau [50] and Boyer et al. [51]. In these terms, the discrete variation of refraction indexes between flow components (i.e., air and water) allows phase discrimination since, at a given emission of light, the amount reflected by the wall of an optical probe sensitive tip depends exclusively on the refraction index of the medium that surrounds the wall. Hence, the quantity of light reflected is received as an optical signal, which is then converted into an electrical signal by a photo-sensitive element. Once this is done, the void fraction is obtained as the portion of time in which the gas phase is contacting the sensitive tip of the optical fiber probe, in relation with the full observation time.

## 3. Results and Discussion

A first analysis of the performance of the numerical and the physical model carried out over basic variables is summarized in Table 3.

**Table 3.** Basic flow variables for the numerical and physical models.

| Model | Supercritical Flow Depth ($y_1$) | Subcritical Flow Depth ($y_2$) | Unit Discharge ($q$) | Inflow Froude Number ($Fr_1$) |
|---|---|---|---|---|
| Numerical model | 1.520 m | 9.500 m | 29.143 m$^2$/s | 4.97 |
| Physical model | 0.061 (1.525) [1] m | 0.370 (9.250) [1] m | 0.233 (29.143) [1] m$^2$/s | 4.93 |

[1] In parenthesis: values at prototype scale applying the scale factor.

The results in Table 3 show that both models were able to reproduce the case study with a similar Froude number for the flow entering the stilling basin. For comparison purposes with bibliographical information, a value of $Fr_1$ = 4.95 is considered for the numerical and physical models.

### 3.1. Free Surface Profile

An analysis of flow depths and the free surface profile was conducted for both, the numerical and the physical model. Results were contrasted and compared between them and also with bibliographical data and expressions in the literature.

The sequent depth ratio is obtained as the ratio between the downstream ($y_2$) and the upstream ($y_1$) flow depth to the hydraulic jump. According to the values shown in Table 3, the sequent depth ratios were 6.25 and 6.07 for the numerical and the physical model, respectively. The sequent depth

ratio was also calculated with the expression proposed by Hager and Bremen [52], based on Bélanger's equation [53], which was developed for the particular case of a classical hydraulic jump (CHJ). The result obtained is 6.30. Therefore, both models provide slightly lower ratios. With reference to the bibliographical value, the numerical model has an accuracy of 99.2% and the physical model, 96.2%. It should be remarked, though, that these lower values of sequent depth ratios are in good agreement with the research presented by Hager and Li [54] and Padulano et al. [30] regarding USBR II stilling basins.

The hydraulic jump efficiency ($\eta$), which gives a measure of the flow energy dissipated by the hydraulic jump [55], is calculated using the expression:

$$\eta = \frac{H_{01} - H_{02}}{H_{01}} \tag{12}$$

where $H_{01}$ and $H_{02}$ are the specific energy heads upstream and downstream of the hydraulic jump. This efficiency is 0.505 for the numerical model and 0.516 for the physical model, whereas using the sequent depth ratio previously calculated for the CHJ, the resulting efficiency is 0.501. Hence, accuracy above the 97% is achieved for both models with respect to the bibliographical values. The resulting efficiency from both models is slightly over the value calculated for a CHJ. This result reflects the specific design characteristics of the stilling basin to improve energy dissipation in the hydraulic jump, and is in good agreement with previous research [30].

Therefore, for both models, the sequent depth ratio is lower than the one for a CHJ, while the efficiency is higher. These results were expected as a consequence of the energy dissipation devices existing in the USBR II stilling basin and, as mentioned before, are in good agreement with previous studies. However, the differences found were not as significant as the ones reported by Padulano et al. [30], probably due to the higher $Fr_1$ values used in [30] with respect to the ones above mentioned.

The dimensionless free surface profile was obtained for both the CFD and the experimental model and is displayed in Figure 4, altogether with data from other authors [8,56]. The dimensionless profile is calculated following the expressions by Hager [55]:

$$X = \frac{x - x_0}{L_r} \tag{13}$$

$$Y = \frac{y - y_1}{y_2 - y_1} \tag{14}$$

where $x_0$ is the hydraulic jump toe position and $L_r$ is the hydraulic jump roller length, according to Hager et al. [57]:

$$L_r = y_1\left[-12 + 100 tgh\left(\frac{Fr_1}{12.5}\right)\right] \tag{15}$$

It can be observed that the models were able to reproduce satisfactorily the hydraulic jump free surface profile, as their profiles are close to the ones previously reported by Bakhmeteff and Matzke [56] and Wang and Chanson [8] for a CHJ. Using the coefficient of determination $R^2$ [58] as a measure of the accuracy of the modeled profiles, the FLOW-3D model achieves $R^2$ values of 0.979 and 0.977 when compared with Bakhmeteff and Matzke [56] and Wang and Chanson [8], respectively. For the physical model $R^2$ is 0.937 when compared to Bakhmeteff and Matzke [56] and 0.944 in relation to the profile by Wang and Chanson [8].

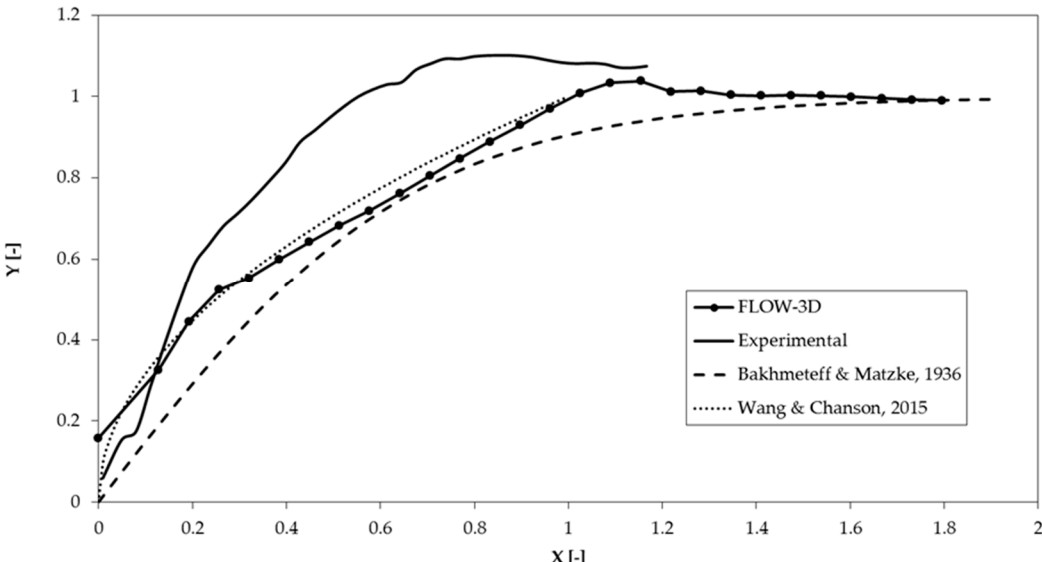

**Figure 4.** Dimensionless free surface profile of the hydraulic jump performed by the numerical and physical models, and contrast with data by Bakhmeteff and Matzke [56] and Wang and Chanson [8].

It should be noticed that near the origin ($X = 0$), that is, close to the hydraulic jump toe position, there is an overestimation of the profiles as obtained in the present research, when compared to results for the CHJ. This is even more significant for the central part of the roller, up to $X = 1$, especially for the profile derived from the physical model. The main reason explaining these differences is the affection of the energy dissipation devices in the properties of the specific hydraulic jump under study. These include the chute blocks existing in the upstream part of the stilling basin and the ones at the end sill. Moreover, the fact of not using a measured value of $L_r$ in Equation (13) affects the estimated dimensionless profile. As stated before, the theoretical value of $L_r$ (Equation (15)) is used to define $X$ in Equation (13). The use of this theoretical value particularly affects the dimensionless profile derived from the physical model, as it showed to be more sensitive to the effect of energy dissipation devices (lower sequent depth ratio and higher dissipation efficiency than the CFD model). This fact could explain the larger deviations shown in Figure 4 for the experimental profile.

Concerning profiles closer to the end of the basin ($X > 1$), flow depths clearly tend to decrease, as predicted for the CHJ in [56]. These results are in agreement with other numerical approaches recently published [3].

### 3.2. Velocity Profiles

Velocity in the streamwise direction was measured in three different sections along the presented model. Firstly, a section in the spillway was considered, where supercritical flow with an almost uniform velocity distribution was obtained for both models. The following two locations were chosen right upstream and downstream of the end sill. As previously explained in Section 2.2.2, the latter choice prevents the use of the turbine velocity meter device in the roller zone, where severe velocity fluctuations take place. For comparison purposes, velocity profiles from the numerical model were also obtained for the same three chosen sections.

Figure 5 compares such profiles upstream and downstream the end sill, resulting from both models. The effect of the energy dissipation device can be clearly observed in the velocity profiles. In particular, and as a result of the sill interference with the flow, the velocity magnitude decreases in the lower part of the profile. As there is not a significant variation in the flow depth for these two relatively close profiles, such a velocity decrease in the lower part is necessarily linked to a velocity increase in the upper part of the profile, as shown in Figure 5.

Since the analyzed profiles were taken downstream the hydraulic jump roller, they do not strictly match other velocity distributions obtained for the internal flow in hydraulic jumps [59]. Some differences are also found with other velocity profiles proposed in contributions devoted to free surface flow in open channels, i.e., Kirkgoz and Ardiclioglu [60]. These differences are mainly due to the affection of the energy dissipation devices on the hydraulic jump under study.

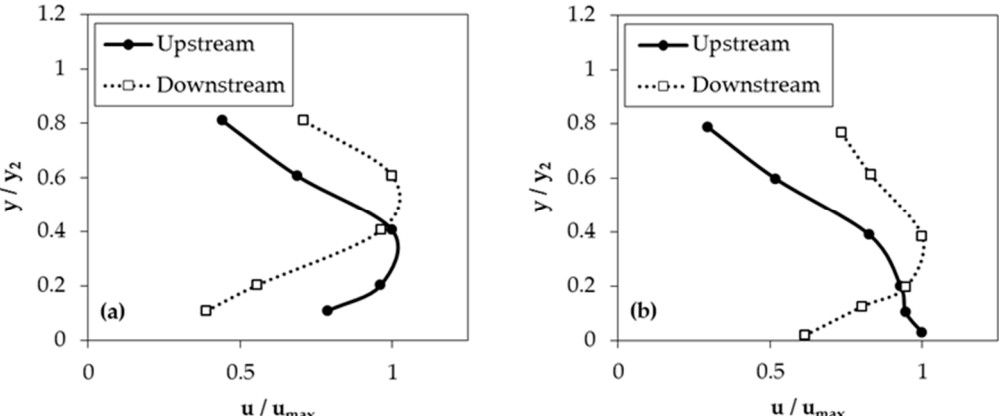

**Figure 5.** Vertical profiles of streamwise normalized velocity measured upstream and downstream the end sill of the USBR II stilling basin: (**a**) Physical model, (**b**) numerical model.

*3.3. Pressure Analysis*

Pressure values were measured in two different positions on the front side of two of the blocks forming the end sill of the USBR II stilling basin. It is important to highlight that the measures taken in both blocks provided similar pressure values. Averaged relative pressure in the end sill was obtained as $p/(\rho g y_1)$. The resulting values for both, the numerical and physical models, range between 5 and 6, whereas, for that same relative position in a CHJ, Toso and Bowers [28] reported values around 6.5. These lower relative pressure values are associated with specific properties of the hydraulic jump in the USBR II, already mentioned in Section 3.1, in particular, the efficiency and the lower sequent depth ratio if compared to that of the CHJ.

Regarding the drag force acting on the sill, it can be expressed by means of the drag coefficient ($C_d$). Padulano et al. [30] proposed the following expression to obtain the drag coefficient in a USBR II stilling basin:

$$C_d = \frac{1 - (1+S)^2\left(\frac{y_2^*}{y_1}\right)^2 - 2Fr_1^2\left\{\left[(1+S)\left(\frac{y_2^*}{y_1}\right)\right]^{-1} - 1\right\}}{\lambda\frac{d_2}{y_1}Fr_1^2} \tag{16}$$

where $S = \left(y_2 - y_2^*\right)/y_2^*$ is the submergence factor, $y_2^*$ the subcritical flow depth calculated using Bélanger's equation [53], $\lambda$ is the ratio of blocked width to total basin width, and $d_2$ is the height of the end sill (Figure 1). The drag coefficients resulting from the numerical and the physical models using this expression are 0.15 and 0.26, respectively. These values are lower than those observed by Padulano et al. [30]. The already mentioned differences in sequent depth ratios with respect to those reported in [30] explain these lower values of the drag coefficient. An alternative expression for the drag coefficient can be found in Hager [55]:

$$C_d = \frac{F_D}{\frac{1}{2}\rho d_2 B u_1^2} \tag{17}$$

where $B$ is the stilling basin width and $F_D$ the drag force. Equation (17) can be used instead of Equation (16) to estimate $C_d$ coefficient, as long as the drag force value is available. If $F_D$ is estimated

as the product of the measured pressure and the vertical projection of the sill area, resulting $C_d$ values are 0.35 for the FLOW-3D model and 0.71 for the experimental model.

### 3.4. Void Fraction Distribution

### 3.4.1. Theoretical Development

This section deals with the analysis of the void fraction distribution throughout the hydraulic jump taking place in the USBR II stilling basin models. As in previous sections, results are compared to those of a CHJ. To do so, it is important to unify criteria and work with normalized expressions. Hence, the void fraction analysis was undertaken on the basis of the expressions proposed by Murzyn et al. [18]. This formulation differentiates two flow regions in order to model the void fraction vertical profiles throughout the hydraulic jump, namely, the lower and the upper region. They are separated by the turbulent shear layer. On the one hand, for the lower region in terms of flow depth, the void fraction ($C$) distribution follows a diffusion equation [61], leading to the profile expression:

$$C = C_{max} exp\left[ -\frac{1}{4} \frac{u_1}{D} \frac{(\xi - \xi_{Cmax})^2}{x} \right] \tag{18}$$

where $D$ is a diffusion coefficient, $\xi$ is the normalized depth obtained as $y/y_1$ and $\xi_{Cmax}$ is the normalized flow depth at which the void fraction reaches its maximum ($C_{max}$). On the other hand, conditions similar to the edge of water jets freely discharging into the air are assumed for the upper region and, accordingly, the expression by Brattberg et al. [62] is proposed:

$$C = \frac{1}{2}\left[ 1 + erf\left( \frac{\xi - \xi_{C50}}{2\sqrt{Dx/u_1}} \right) \right] \tag{19}$$

where a void fraction of 0.5 is reached at $\xi_{C50}$. It should be noticed that $u_1$ value (flow velocity upstream the hydraulic jump toe) is the same for both Equations (18) and (19). However, the diffusion coefficient value varies between Equations (18) and (19). This is due to the different air entrainment mechanisms occurring in the two defined regions.

Expressions in Equations (18) and (19) were fitted to the laboratory measurements taken in selected sections along the stilling basin. The same procedure was carried out using void fraction values, drown from the numerical model [9].

The initial analysis showed a high level of coincidence between $\xi_{Cmax}$ and the normalized depth of the boundary between regions ($\xi_*$). If $\xi_{Cmax}$ is set equal to $\xi_*$ the two distributions are automatically linked by this boundary common value, and the approach becomes coherent. In practice, this implies a more parsimonious model representing the whole void fraction profile.

### 3.4.2. Void Fraction Analysis—Case study

Six different profiles were considered to characterize the void fraction distribution along the hydraulic jump longitudinal axis. Table 4 shows the normalized x-position of these profiles for the presented models.

**Table 4.** Location of the vertical profiles measured for the void fraction distribution analysis.

| Model | $x/y_1$ | | | | | |
|---|---|---|---|---|---|---|
| Numerical model | 1.32 | 5.84 | 11.18 | 20.64 | 26.56 | 33.96 |
| Physical model | 1.31 | 5.82 | 11.14 | 20.57 | 26.47 | 33.85 |

Figure 6 shows void fraction values obtained for a selected section (Table 4), drawn from laboratory measurements (Figure 6a), and numerical results (Figure 6b). The fitted theoretical distributions according to Equations (18) and (19) are also shown in these graphs.

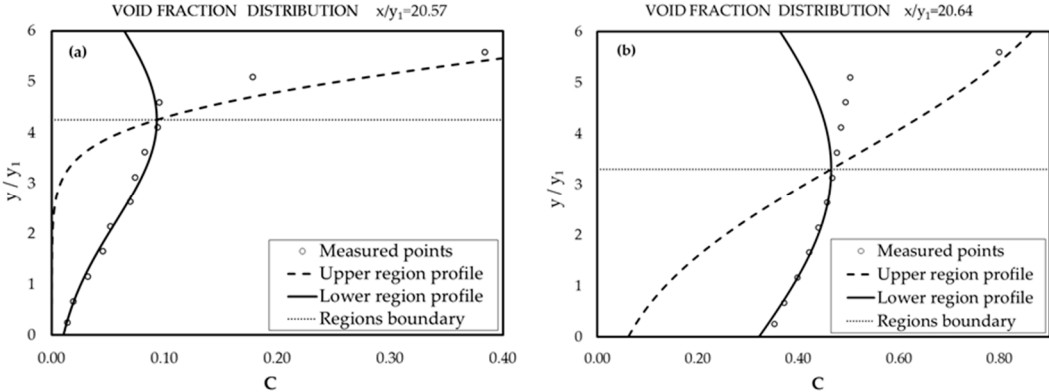

**Figure 6.** Measured void fraction in a vertical profile with the adjusted expressions for the upper and the lower regions: (**a**) Physical model ($x/y_1$ = 20.57), (**b**) numerical model ($x/y_1$ = 20.64).

The analysis of the data obtained from the models and the adjustment process of the theoretical profiles previously presented, led to the exclusion of part of the data from the analysis. Firstly, the numerical model showed that values corresponding to the upper region of the flow did not follow the distribution presented in Equation (19). Therefore, the aeration mechanism occurring in this region is not adequately reproduced by the model (Figure 6b). In fact, flow aeration in CFD modeling remains as one of its main challenges [7,9,18]. Highly aerated flows, like the one here presented, remark the limitations of the VOF approach. Moreover, in this particular case, modeling aeration involves the estimation of several parameters by means of a thorough calibration process. Hence, in the present research, the numerical model was not considered for the upper region analysis.

Regarding experimental measures, the data obtained with the optical fiber probe in the first profile ($x/y_1$ = 1.31) showed an anomalous behavior, which is not observed in subsequent sections. This can be explained by the proximity to the hydraulic jump toe. The relatively low flow depth for this profile, together with the presence of intense turbulent fluctuations, may have affected the performance of the probe. Consequently, this first profile was excluded from the analysis concerning physical model data.

Using the refined data sample, estimated parameters of Equations (18) and (19) were analyzed for both, the upper and the lower region. Results are then compared with data from Murzyn et al. [18] and Chanson and Brattberg [17], obtained for a CHJ of $Fr_1$ 3.7 and 6.3, respectively (Figures 7 and 8).

Regarding the parameter $C_{max}$, that is, maximum void fraction in the lower region, Murzyn et al. [18] suggested an expression of the form:

$$C_{max} = \alpha \times exp(-Ax/y_1) \qquad (20)$$

where $\alpha$ and $A$ are parameters. For the physical model, the estimated values were $\alpha$ = 0.35 and $A$ = 0.07, whereas for the numerical model these estimated parameters were $\alpha$ = 0.65 and $A$ = 0.02. Equation (20) provides a very good representation of $C_{max}$ variation along the hydraulic jump (Figure 7a). $R^2$ values are 0.966 and 0.928 for the physical and numerical model, respectively. For the physical model, the rate of decrease for $C_{max}$ is very close to those reported by Murzyn et al. [18] and Chanson and Brattberg [17]. For the numerical model, though, such a rate is slightly slower, as shown in Figure 7a.

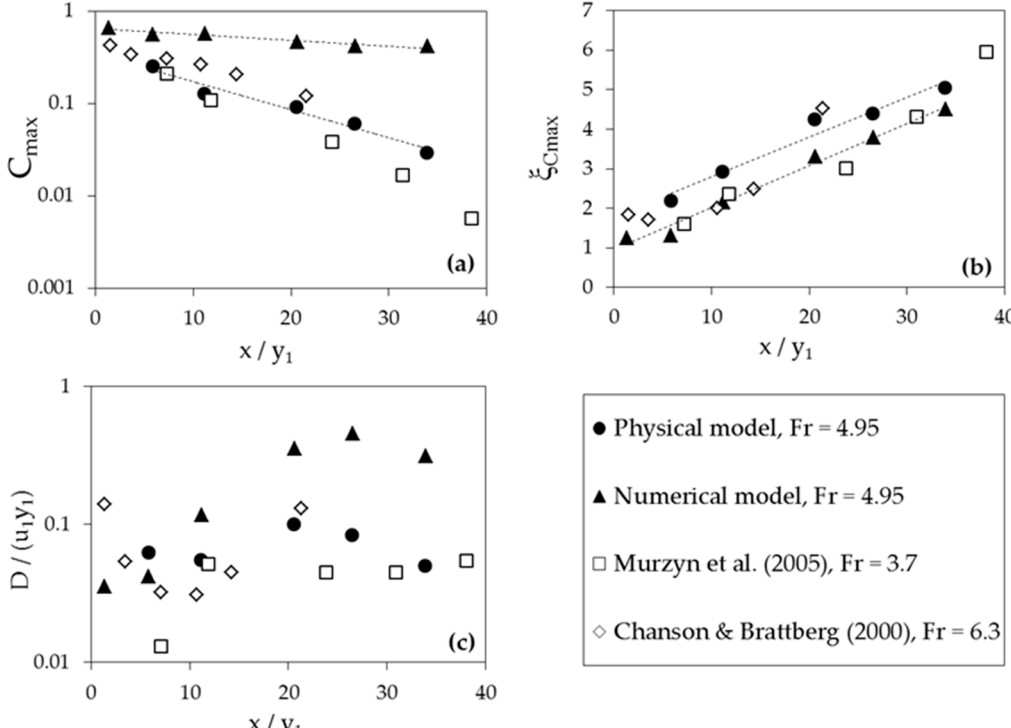

**Figure 7.** Parameters of the void fraction distribution in the hydraulic jump (lower region): (**a**) Maximum void fraction ($C_{max}$), (**b**) normalized height for the maximum void fraction ($\xi_{Cmax}$), (**c**) diffusion coefficient ($D$).

Figure 7b shows results concerning the normalized height at which this maximum void fraction is reached ($\xi_{Cmax}$). Clearly $\xi_{Cmax}$ increases as the distance to the hydraulic jump toe increases. The estimated gradient was 0.101 (physical model) and 0.107 (numerical model). These results are almost coincident to those obtained by Murzyn et al. [18] and Chanson and Brattberg [17], reporting values of 0.102 and 0.108, respectively.

In respect to the estimated values of the diffusion coefficient ($D$) for the lower region (Figure 7c), results from the present research yielded to generally higher $D$ values, as compared to previous research in Murzyn et al. [18] and Chanson and Brattberg [17]. A significant dispersion of $D$ values is observed, in line with Murzyn et al. [18] previous observations.

In conclusion, estimated parameters for the lower region show a satisfactory agreement with previous values reported in the literature. Nevertheless, it should be noticed that the rate of decrease in $C_{max}$ found was slightly lower as compared to bibliographical data. In particular, for void fraction distributions drown from the numerical model. On the other hand, relatively higher $\xi_{Cmax}$ were also found herein, when compared to other sources. Both differences could be a consequence of the energy dissipation devices of the USBR II stilling basin, causing affection to the aeration structure when compared to the CHJ case. This effect has been previously reported by the authors [9], leading to higher void fraction values. This observation is also in good agreement with the results pointing out an increase in flow depths for the hydraulic jump roller, already discussed in Section 3.1.

Figure 8 presents results obtained for estimated parameters characterizing void fraction distributions in the upper region. Apart from the diffusion coefficient ($D$) and the normalized height of the boundary between regions ($\xi_*$), the normalized heights at which $C$ is 0.95 ($\xi_{C95}$) and 0.5 ($\xi_{C50}$) were investigated. As mentioned before, only data from the physical model were analyzed for this upper region, discarding numerical model results. As shown in Figure 8, there is a general good agreement with previous contributions [17,18]. All three parameters $\xi_*$, $\xi_{C95}$, and $\xi_{C50}$ presented increasing values as we departed from the hydraulic jump toe. It is interesting to remark that the rate

of increase for parameters $\xi_*$, $\xi_{C95}$, and $\xi_{C50}$ was higher for sections closer to the hydraulic jump toe, and not so significant for further locations, as it is clearly shown in Figure 8a–c. These observations showed that the region where interfacial aeration is the predominant air entrainment mechanism increases its thickness with the distance to the hydraulic jump toe. This behavior is also pointed out by Murzyn et al. [18]. Hence, the change in the rate of increase of these normalized heights is associated with a widening of the upper region.

With regards to the diffusion coefficient values (Figure 8d), a decreasing trend with x could be identified in the upper region, unlike previously estimated values for the lower region. This is in good agreement with the observations made by Murzyn et al. [18] and Chanson and Brattberg [17].

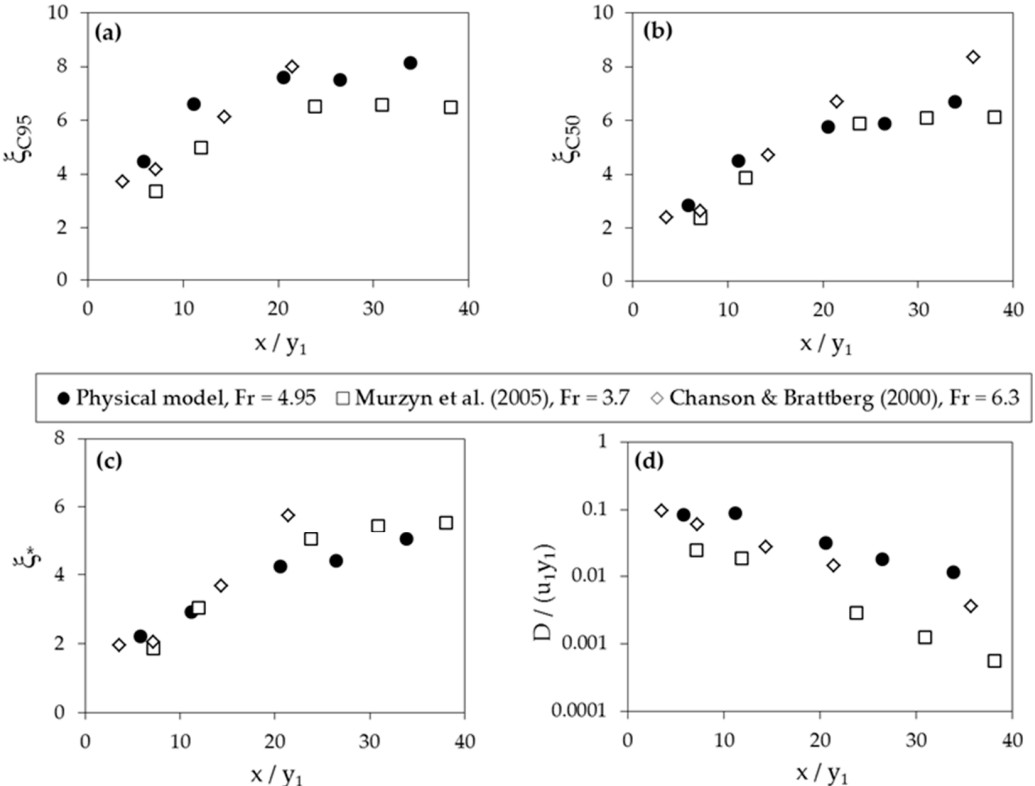

**Figure 8.** Parameters of the void fraction distribution in the hydraulic jump (upper region): (**a**) Normalized height at which the void fraction is 0.95 ($\xi_{C95}$), (**b**) normalized height at which the void fraction is 0.5 ($\xi_{C0.5}$), (**c**) normalized height of the boundary between regions ($\xi_*$), (**d**) diffusion coefficient (*D*).

## 4. Conclusions

A general and representative case study consisting of a Creager spillway and a typified USBR II stilling basin was analyzed using a reduced-scale physical model and a CFD numerical model. The results obtained in the experimental campaign and in the processing of the FLOW-3D simulations were compared, not only between them but also with data and expressions coming from an extensive literature review. The objective of this comparison was to assess the performance of the physical and the numerical models and to characterize the hydraulic jump taking place in the stilling basin. In particular, identifying the effect of the energy dissipation devices on these hydraulic jump structural properties, as compared to a classical hydraulic jump, was intended.

The analysis of the results showed that both models were able to adequately represent the flow in the spillway and the stilling basin. In terms of the hydraulic jump, its sequent depth ratio, efficiency, and free surface profile were successfully reproduced by the numerical and the physical models. A satisfactory agreement was found with previous results. Nevertheless, a slight underestimation of the

affection of the energy dissipation devices to the flow is reported. Both models developed herein had the ability to adequately reproduce velocity profiles. These results revealed the important influence of the dentated sill placed at the end of the USBR II stilling basin. The analysis of the pressure distributions showed results in line with previous bibliographical observations, although the drag coefficient tended to be underestimated by the presented models when compared to other authors' contributions.

Finally, for the void fraction distribution, an extensive analysis was carried out, based on a reformulation of previous void fraction profile theoretical expressions. This analysis showed that the optical fiber probe measurements provide a very good representation of the aeration process. Resulting void fraction profiles showed a high degree of coincidence with observations made by other authors. Despite these similarities, the model was able to reveal the influence on the flow of the energy dissipation devices in the stilling basin, not present in the CHJ flow. The FLOW-3D numerical model showed acceptable results for the lower region but was not able to reproduce the aeration mechanisms occurring in the upper region, where interactions with the free surface dominate.

The models presented herein have several limitations that should guide further research on the topic. Firstly, an extended experimental campaign with new instrumentation would improve velocity profile representation, pressure distributions, and air entrainment. This will enhance calibration of the numerical model parameters and coefficients. It will provide a better understanding of the affection of the energy dissipation devices to the hydraulic jump characteristics. Furthermore, the case study proposed was referred only to a particular hydraulic design, with flow occurring with $Fr_1 = 4.95$. This case study could obviously be redesigned, changing the flow conditions and consequently adapting the structure to cover a wider range of Froude numbers. This will certainly provide a wider perspective of the possible hydraulic jump types taking place and their properties.

Nonetheless, both the physical and the numerical model showed good performance, being able to adequately reproduce the flow under study. Hence, this work encourages the use of methodologies based on a double numerical and physical modeling approach to study complex flows in hydraulic structures. In particular, the results reported herein contribute to enhancing the knowledge of the flow in a typified USBR II stilling basin, which, in turn, can be used to improve their design. This is a key issue in hydraulic engineering due to the increasing importance of existing stilling basins adaptation to higher discharges than those considered in their original design. The issue becomes even more relevant under the new society demands in terms of new dam safety requirements in the framework of climate change scenarios.

**Author Contributions:** The conceptualization of this research was directed by F.J.V.-M., who also assisted in the experimental analysis altogether with B.H. and A.B. The experimental campaign was carried out by J.F.M.-P. who also developed the numerical model with the supervision of B.H. and A.B. The analysis and discussion of the results were done by R.G.-B. and J.F.M.-P. Drafting of the document was done by R.G.-B. and J.F.M.-P. with contributions by the rest of the authors. All authors have read and agreed to the published version of the manuscript.

**Funding:** This research was funded by 'Generalitat Valenciana predoctoral grants (Grant number [2015/7521])', in collaboration with the European Social Funds and by the research project: 'La aireación del flujo y su implementación en prototipo para la mejora de la disipación de energía de la lámina vertiente por resalto hidráulico en distintos tipos de presas' (BIA2017-85412-C2-1-R), funded by the Spanish Ministry of Economy.

**Acknowledgments:** The authors would like to acknowledge the collaboration of the Hydraulics Laboratory of the Institute of Hydraulic Engineering and Water Resources Management from Technische Universität Wien (TU Wien) and their technicians, in the construction and experimental campaign of the physical model referred in the article.

**Conflicts of Interest:** The authors declare no conflict of interest. The funders had no role in the design of the study; in the collection, analyses, or interpretation of data; in the writing of the manuscript, or in the decision to publish the results.

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
