# Peer review of "Analysis of the Flow in a Typified USBR II Stilling Basin through a Numerical and Physical Modeling Approach"

_water, doi:10.3390/w12010227_

Round 1

Reviewer 1 Report

I read with interest the paper “Analysis of the Flow in a Typified USBR II Stilling Basin through a Numerical and Physical Modeling Approach”, and I feel it is interesting and relevant for the readers. The use of joint experimental and numerical analyses to deal with complex hydraulic structures is definitely useful for the scientific community and potentially also for some practitioners. The Authors also deal with a really challenging issue which is the flow aeration. The design of the research is solid and appropriate, and the paper clear and well written. Although some results could be improved, I feel that the paper deserves to be published once some issues are addressed:

Please revise carefully the language. There are some typos throughout the text that should be corrected.

A question on the boundary conditions used for the CFD model is whether additional BC were used in some simulations or not. Testing additional flow conditions could be a value added of a validated numerical model. Did the authors set any Initial Condition?

A summary (table of figure) with the results of the mesh sensitivity analysis would be useful for the reader.

Considering the physical model, some considerations on the uncertainty associated to the parameters that are measured should be included.

Maybe the Figure 4 is misleading, but apparently the physical model provides quite different results compared to the other data. Particularly for x ranging between 0.2 and 1, the value of Y is significantly different due, according to the Authors, to the effect of dissipating structures. Conversely, the numerical model is in good agreement with literature data. This aspect needs further clarification since, apparently, different ‘conditions’.

Concerning results and discussion, I would suggest to go further into details even into the most critical issues, such as flow aeration. Would it be possible, e.g. to check how the model performs in different flow conditions? How could it be improved, particularly concerning the numerical aspects?

Reviewer 2 Report

The paper is of interesting and important topic and focuses on the problem of adaptation of stilling basins to higher discharges than those considered for their design. I particular the Authors presents modelling based approach to analyzing the hydraulic jump occuring in an USBR III stilling basin.

The paper is of good structure and was prepared with necessary care. The theoretical background properly illustrates the problem that is analyzed in the paper. The model was described properly and provided numerical experiments (based on physical model) were widely disscussed.

In my opinion - the paper is a really good one (it has good scientific value and was prepared with necessary care). Therefore I suggest to publish it at present form. Congratulations!
